# Behavioural Responses of Male *Aedes albopictus* to Different Volatile Chemical Compounds

**DOI:** 10.3390/insects13030290

**Published:** 2022-03-15

**Authors:** Davide Carraretto, Laura Soresinetti, Irene Rossi, Anna R. Malacrida, Giuliano Gasperi, Ludvik M. Gomulski

**Affiliations:** Department of Biology and Biotechnology, University of Pavia, 27100 Pavia, Italy; davide.carraretto01@universitadipavia.it (D.C.); laura.soresinetti01@universitadipavia.it (L.S.); irene.rossi03@universitadipavia.it (I.R.); malacrid@unipv.it (A.R.M.); gasperi@unipv.it (G.G.)

**Keywords:** *Aedes albopictus*, Asian tiger mosquito, olfactometer, attractant, repellent, odorants, trapping

## Abstract

**Simple Summary:**

Many studies have been performed to assess the effects of chemical compounds on mosquito behaviour. These studies almost exclusively involve only female mosquitoes as they can transmit disease pathogens, or at least, cause biting nuisance. Few studies have considered male mosquitoes. The identification of chemical substances that attract males can be very useful for trapping purposes, especially for monitoring the makeup of the male population during control programmes, such as those involving the release of sterile male mosquitoes. Twenty-eight chemical compounds from different chemical classes were evaluated using a dual-port olfactometer assay with at least three serial hexane dilutions against a hexane control. The compounds included known animal, plant and fungal volatiles, and the components of a putative *Aedes aegypti* pheromone. Many of the compounds were repellent for male mosquitoes, especially at the highest concentration. One compound, decanoic acid, acted as an attractant for males at an intermediate concentration. Decanoic acid did not elicit a significant response from female mosquitoes.

**Abstract:**

The Asian tiger mosquito, *Aedes albopictus*, has become one of the most important invasive vectors for disease pathogens such as the viruses that cause chikungunya and dengue. Given the medical importance of this disease vector, a number of control programmes involving the use of the sterile insect technique (SIT) have been proposed. The identification of chemical compounds that attract males can be very useful for trapping purposes, especially for monitoring the makeup of the male population during control programmes, such as those involving the use of the SIT. Twenty-eight chemical compounds from different chemical classes were evaluated using a dual-port olfactometer assay. The compounds included known animal, fungal and plant host volatiles, and components of a putative *Aedes aegypti* pheromone. Many of the compounds were repellent for male mosquitoes, especially at the highest concentration. One compound, decanoic acid, acted as an attractant for males at an intermediate concentration. Decanoic acid did not elicit a significant response from female mosquitoes.

## 1. Introduction

The study of the chemically-mediated behaviour of male mosquitoes has, to a large extent, been ignored in favour of the more exciting animal host odour-related behaviour of their blood-feeding female counterparts [1]. Male mosquitoes, like females, require nectar as a source of carbohydrates and respond to semiochemicals released by suitable host plants. Semiochemicals may also signal the presence of suitable resting sites. As mosquitoes are thought to be important in the pollination of certain plant species [2,3], it is highly likely that coevolution between a mosquito species and a flowering plant species will have resulted in plant-mosquito specialisations. Certain plant volatiles may therefore be expected to be attractive or repellent for some mosquito species but not others. For example, eugenol has been shown to be repellent for *Ae. Aegypti* and *Culex quinquefasciatus*, but not *Anopheles coluzzi* [4]. Within a species, males and females appear to respond to, and share, the same preferences for host plants [5,6]. Many mosquitoes species have preferred animal hosts, so certain host emanations may be mosquito species-specific, but perhaps to a lesser extent than plant volatiles. Certain host cues, such as carbon dioxide, however, appear to be important for all blood-feeding species [7]. Males may also respond to the same animal host odours as blood-seeking females, as these represent resources where sexually receptive females might be found [8,9]. The identification of the semiochemicals responsible for behavioural responses (i.e., attraction to plant and animal hosts) could lead to new and effective synthetic attractants that target both male and female mosquitoes. Such attractants could be used in traps for surveillance, monitoring the success of control programmes, or for lure and kill approaches [10,11].

Here we used a dual-port olfactometer to test the responses of male *Ae. albopictus* to different concentrations of twenty-eight chemical compounds that constitute components of plant, fungi, human and animal host emanations, and two putative components of an *Ae. aegypti* aggregation pheromone [12]. The compounds belong to the following range of different chemical classes: ketones, terpenes, esters, aldehydes, alcohols and a fatty acid. Many of these compounds have been previously implicated in male and female mosquito behavioural responses [1,10,11,13,14], or have been shown to activate odour receptors in different mosquito species [15,16]. Our hypothesis was that male mosquitoes may be attracted to one or more concentrations of these compounds.

## 2. Materials and Methods

### 2.1. Insects

Six to ten day old male and female *Ae. albopictus* individuals of the Rimini strain were used in all our experiments. This strain was established by Romeo Bellini of the Centro Agricoltura Ambiente “Giorgio Nicoli” in 2004 from eggs collected at Rimini, Emilia Romagna, Italy, and has been maintained in Pavia since 2013. Larvae were reared on fish food (Tetra Goldfish granules, Tetra GmbH, Melle, Germany). Pupae were collected daily in a small volume of rearing water in small plastic cups and placed in an empty 20 cm × 20 cm × 20 cm cage (Bugdorm, NHBS, Totnes, UK). Newly emerged adults were collected daily. Male and females were maintained together in cages (sex ratio approximately 1:1). Adults were given access to 20% glucose solution and were maintained at 26 °C and 65–70% relative humidity under a 12:12 h (light:dark) photoperiod. The dark period was from 20:00 to 8:00. The females used in our assays were not blood-fed.

### 2.2. Olfactometer

A dual-port olfactometer was used to determine the responses of *Ae. albopictus* to different stimuli (odorants). The olfactometer was constructed using plexiglass (Figure 1). The airstream (medical-grade air source) was filtered using activated carbon and then humidified using a bubble bottle containing distilled water. The airstream was then split into two streams, each controlled by flow meters (0–10 L/min). The streams passed into the two arms of the olfactometer. Conical funnels were positioned within the arms leading to traps whose distal ends were covered in net fabric (Figure 1). The two arms were designed to accommodate a plexiglass cylinder that could contain either live individuals or other stimulants (see below). One end of the cylinder was covered in net fabric while the other end was covered by netting held in place by an elastic band. Alternatively, the arms could accommodate chemical compounds, as described in Section 2.4.

### 2.3. Olfactometer Validation

To validate the olfactometer (Appendix A), a known mosquito attractant, human foot odour [17], was used. A polyamide sock (Ciak 15 Sanpellegrino, CSP International Fashion Group SpA, Ceresara, Italy) was worn by one of the male authors (LMG) for three days and nights to accumulate foot odour [18,19]. The worn sock (treatment) was placed in a plexiglass cylinder and closed at each end with net fabric. An unworn sock (control) was placed in an identical cylinder. The two cylinders were positioned in the distal part of the arms of the olfactometer. The entire olfactometer was covered by a white cotton sheet to avoid visual stimulation of the mosquitoes during the test. The olfactometer was illuminated from above the sheet by a 60 watt halogen light bulb (OSRAM GmbH, Munich, Germany). The illuminance just above the olfactometer was around 2000 lux. Twenty *Ae. albopictus* individuals (males or females) were released into the main chamber of the olfactometer by inserting the release chamber and rotating it 180 degrees, thus allowing the plexiglass lid to swing open, permitting the mosquitoes to enter the main chamber.

The end caps of the two arms were then fixed in position and the airflow commenced, and maintained, at a constant rate of 3 L/min. After 10 min had elapsed, the number of individuals in the traps of the two arms were counted. The cylinders containing the socks were removed and all of the mosquitoes were removed and eliminated after each run. Clean air was allowed to flow through the olfactometer for 15 min between replicates. The position of the worn and control socks in each arm was alternated for each replicate. Five replicates were performed for each sex. Given that the number of individuals trapped in each arm may not have a normal distribution, and that they could be considered proportions, each number was arcsine transformed [20] and compared using a paired, two tailed *t* test. This assay was always conducted between 15:00 and 19:00 to limit the influence of circadian factors. The temperature within the olfactometer was 27 ± 0.72 °C with 60.8 ± 4.0 relative humidity (RH), as determined by a thermometer-hygrometer (Tacklife HM01).

### 2.4. Chemical Compounds and Preparation

The chemical compounds (Sigma-Aldrich/Merck Life Science Srl, Milan, Italy) listed in Table 1 were tested against *Ae. albopictus* for a response.

At least three serial dilutions in hexane (>95%, Merck) were tested for each compound (1% by volume, or mass in the case of solids (decanoic acid), 10^−2^; 0.01%, 10^−4^; 0.0001%, 10^−6^). An additional dilution of 0.000001%, 10^−8^, was tested for 4-propyl-benzaldehyde.

A 400 µL aliquot of the hexane dilution of the test compound, or 400 µL of hexane (control), was spotted on a 20 mm × 35 mm Whatman 3MM filter paper rectangle supported by a 24 mm× 40 mm glass microscope coverslip. The hexane was allowed to evaporate for 2 min (under a benchtop extractor arm) and then the control and treatment filters/coverslips were positioned in each arm of the olfactometer on an inverted 2 cm thick solid watch glass (Figure 1). The olfactometer was covered by a white cotton sheet and illuminated from above as described in the previous section. Twenty *Ae. albopictus* males were released into the main chamber of the olfactometer, and after 10 min had elapsed, the number of individuals in the trap from each arm was counted. The filter papers with supporting microscope coverslips, and all the mosquitoes, were removed and eliminated after each run. Clean air was allowed to flow through the olfactometer for 15 min between replicates. Compounds were tested with at least six replicate runs for each concentration. The position of the test stimuli and control in each arm was alternated for each replicate. Each compound was initially tested using the low concentration followed by the intermediate and high concentrations. After each set of runs, the apparatus was thoroughly wiped with 70% ethanol using paper towels and air was pumped through the apparatus for 24 h to aid the evaporation and elimination of any contaminating compound. This assay was conducted between 15:00 and 19:00.

We described the effects of the compounds on the mosquitoes in terms of attraction and repulsion: attraction, when a significantly higher number of individuals accumulated in the trap from the arm with the test compound; repulsion, when a significantly higher number accumulated in the control arm. Some compounds might, of course, elicit alternative responses, for example, as an excitant or an arrestant, without attracting or repelling the mosquitoes. We were not able to assess these responses.

## 3. Results

### 3.1. Olfactometer Validation

Female mosquitoes showed a significant preference for the arm of the olfactometer containing the foot odour/worn sock (two tailed paired *t* test, *p* = 0.010). Male mosquitoes did not show any preference and were not attracted to either the foot odour/worn sock or the control sock (Figure 2).

### 3.2. Response of Male Mosquitoes to Chemical Compounds

Seven of the ten ketones tested showed repellency at one or more concentrations (Figure 3b–e,g–i). For 2-nonanone, acetophenone, 6-methyl-5-hepten-2-one, 2,6,6-trimethyl-2-cyclohexanene-1,4-dione, 3,7-dimethyl-oct-6-en-1-yn-3-ol and 4′-ethyl-acetophenone, there was significant repellency at only the high concentration (10^−2^), whereas 4-propyl-benzaldehyde showed repellency at both the intermediate and high concentrations, with greater repellency at the high concentration. For the majority of the assayed ketones, there was an evident, though not always significant, correspondence between the responses of the males and the concentration of the compound. For 4-propyl-benzaldehyde (Figure 3d), there appeared to be a trend of decreasing repellency and increasing attraction for each dilution. For this reason, an additional six replicates of a higher dilution of 10^−8^ were tested, although no significant response was recorded.

Two of the five terpenes tested showed repellency at the 10^−2^ concentration (geraniol, *p* = 0.034; linalool oxide, *p* = 0.030; Figure 3m,o). The other terpenes did not produce a significant response from the males at any concentration.

Of the seven esters tested, two showed repellency, both at the high concentration (methyl benzoate, *p* = 0.007; methyl 2-methylbenzoate, *p* = 0.029; Figure 3r,s).

Two of the three aldehydes tested showed repellency. Nonanal showed significant repellency only at the high concentration (Figure 3w, *p* = 0.025), whereas phenylacetaldehyde showed significant repellency only at the low concentration (Figure 3x, *p* = 0.039).

One of the two alcohols tested, 1-octen-3-ol showed repellency at the high concentration (*p* = 0.025, Figure 3aa).

The only acid tested, decanoic acid, had no significant effect at the low concentration (10^−6^), but at the intermediate concentration (10^−4^), there was significant attraction of males (*p* = 0.025). At the high concentration, the compound acted as a repellent (10^−2^, *p* = 0.027) (Figure 3ab). Given this result, the same serial dilutions of decanoic acid were assayed using female mosquitoes. No significant response to any dilution was recorded (Figure 3ac, Appendix A).

## 4. Discussion

We tested 28 compounds belonging to six chemical classes originating from plants, fungi, or animals, and we provide evidence that 13 acted as repellents at one or more concentrations, usually the highest concentration. Fourteen compounds elicited no significant response, while one compound, decanoic acid, acted as an attractant for males at the intermediate concentration, and as a repellent at the high concentration. Female *Ae. albopictus* showed no significant response to decanoic acid. We discuss our results in relation to those of previous studies, considering for the most part only the mosquito species. This is not a simple task due to a high number of variables, such as different treatment concentrations, test assay conditions, and species and sex of the mosquitoes tested, to mention a few. We have divided the discussion into sections based on the origin of each volatile.

### 4.1. Animal-Related Volatiles

Decanoic acid is a fatty acid that is a surface component of human skin [23] and is present in many plants. It is also a component of mosquito cuticular and internal lipids [24]. In our assay, decanoic acid was found to be attractive at the intermediate concentration and repellent at the high concentration for male *Ae. albopictus*. No effect was observed against females. Decanoic acid acts as a biting deterrent for *Ae. aegypti* with an efficacy similar to that of *N*,*N*-diethyl-*m* toluamide (DEET), but decanoic acid remains effective for a longer period of time [25].

Decanoic acid-treated pools initially became repellent and then subsequently highly attractive for ovipositing females of *Cx. restuans* [26]. The attraction of decanoic acid-treated pools for ovipositing mosquitoes was confirmed in another study (this time for *Cx. pipiens molestus* and *Ae. aegypti*), apparently as a consequence of the attractiveness of the bacterial breakdown products from the decanoic acid substrate [27].

Nonanal, a major odour component of birds and human skin, was repellent at the highest concentration in our tests. At low concentrations, it acts as an attractant for host-seeking mosquitoes (*Ae. aegypti*, *An. gambiae* and *Cx. quinquefasciatus*), while in gravid females it acts as a cue for a suitable ovipositioning site (*An. arabensis*, *Cx. quinquefasciatus*, *Cx. tarsalis*). The same chemical also contributes to the recognition of plant hosts for *Ae. aegypti* [2,10,28,29,30]. The *Plasmodium* parasite induces an increase in the production of certain volatiles by infected individuals, including the aldehydes heptanal, octanal and nonanal. The emission of these volatiles makes humans infected with malaria more attractive to *Anopheles* vectors, resulting in greater transmission of the parasite [31,32].

The alcohol, 1-octen-3-ol, a component of cattle breath and human sweat [33,34], was repellent for *Ae. albopictus* males at the highest concentration we tested. As a lure in traps, it was attractive for females of *An. gambiae*, *Ae. aegypti* and *Ae. albopictus*, but failed to attract *Cx. quinquefasciatus* [35,36] and may be a repellent for the latter species [37].

Finally, 2-nonanone and 6-methyl-5-hepten-2-one (sulcatone) are components of human odour that have been shown to be attractive for *Ae. albopictus* females [36]. In our assays, at the highest concentration, sulcatone acted as a repellent. Indeed, sulcatone appears to act as a “masking”, or even a repellent, odorant for host-seeking *Ae. aegypti* [30,38]. The emission of 6-methyl-5-hepten-2-one from overcrowded or pre-occupied larval sites appears to act as an ovipositioning deterrent for gravid *An. coluzzii* females to reduce the risk of intraspecific competition and cannibalism [39].

### 4.2. Plant-Related Volatiles

Mosquitoes visit flowers for nectar and may, in turn, act as pollinators for the plants [2,40,41]. The interaction between the mosquito and the plant is determined by the composition of the inflorescence odour, which may contain a mixture of attractive and repellant compounds [2]. One such case is provided by the Spanish catchfly, *Silene otites*, whose inflorescences emit a strong odour at night that is attractive for *Culex pipiens molestus* [41]. Individual components of the *S. otites* odour were demonstrated to elicit antennal responses in both male and female *Cx. pipiens* and *Ae. aegypti* [6]. The floral emissions of *S. otites* includes six of the compounds that we tested against male *Ae. albopictus*, namely: 2-phenyl ethanol, phenylacetaldehyde, (*Z*)-3-hexenyl acetate, linalool oxide, acetophenone and methyl salicylate. These compounds were not attractive in our behavioural test; moreover, three of them—phenylacetaldehyde, linalool oxide and acetophenone—were repellent. It is noteworthy that 2-phenyl ethanol, phenylacetaldehyde, linalool oxide and acetophenone were attractive in two choice bioassays for *Cx. p. molestus* females (males were not tested but were equally attracted by the odour of *Silene otites* inflorescences) [6].

A given chemical cue induces behavioural responses by activating some receptors and inhibiting, or having no effect, on others, and this may differ between species [42]. In addition, the concentration of a compound, the assay type and scale of the assay can influence the response. For example, the floral components phenylacetaldehyde and acetophenone, which had previously been shown to be attractive for *Ae. aegypti* in small scale experiments, were not attractive in a larger setting [43]. Moreover, Von Oppen and colleagues [44] found no response from *Ae. aegypti* to 2-phenyl ethanol (at a concentration equivalent to our 10^−4^ dilution) in a Y-tube olfactometer. However, when undiluted 2-phenyl ethanol is applied directly to the skin or clothing, it becomes an extremely effective repellent against female *Ae. aegypti* [45]. Compounds may act as attractants for insects at low concentrations but as repellents at higher concentrations [46]. This is what we observed for decanoic acid.

Methyl cinnamate and methyl salicylate have been identified as the active components of *Ocimum forskolei*, a plant traditionally used in Eritrea as a repellent against mosquitoes, black flies and ticks [47,48,49]. At a concentration of 10^−3^ in hexane, both methyl cinnamate and methyl salicylate significantly reduced landing of *Ae. aegypti* females on human skin odour baits. *Ocimum forskolei*, and other plants in the genus, have been shown to be repellent for other mosquito species such as *An. arabiensis* and *An. stephensi,* suggesting a similar modality of repulsion across mosquito taxa [47]. In our tests, methyl cinnamate was repellent at the lowest concentration whereas methyl salicylate elicited no response at any concentration.

The malarial vector, *An. Gambiae*, has been shown to respond to three of the compounds tested here: R-(+)-limonene, linalool oxide and (E)-β-farnesene [50]. These were among the components isolated from three plants favoured by nectar-feeding *An. Gambiae*: Santa Maria feverfew, *Parthenium hysterophorus*; the castor oil plant *Ricinus communis*; Cobbler’s pegs, *Bidens Pilosa*. We did not record attraction of male *Ae. albopictus* to R-(+)-limonene or farnesene, although we used a mixture of isomers rather than (E)-β-farnesene, and all of our concentrations were higher than the concentrations used against *An. gambiae*. The potential of linalool oxide as a single-component plant-based lure has been investigated for trapping *Aedes* species in unlit bait traps at field sites in Kenya [51]. Linalool oxide-baited traps performed comparably with commercial BioGent (BG) Lure-baited traps for trapping female *Ae. aegypti*, but significantly more males were collected in the linalool oxide traps. When CO_2_ was added, linalool oxide was significantly better than the BG Lure with a 2.8-fold increase in male *Ae. aegypti* captures [51]. In our tests, however, male *Ae. albopictus* were repelled by linalool oxide at our highest concentration.

Hao and colleagues [13] tested several plant volatiles, including geraniol, eugenol and anisaldehyde against *Ae. albopictus* females using a two-port olfactometer. Geraniol and eugenol did not induce a significant response, regardless of concentration, while anisaldehyde elicited significant attraction at a 6% concentration. In our assays, we found no response of male *Ae. albopictus* to eugenol or anisaldehyde. Geraniol did, however, act as a repellent for males at our highest concentration. Geraniol was also shown to be highly repellent for *An. gambiae* females when tested at concentrations between 10^−2^ and 10^−5^ g/mL [52]. Afify and colleagues employed a close proximity assay to demonstrate that 60% eugenol was a repellent for *Ae. aegypti* and *Cx. quinquefasciatus*, but not *An. coluzzii*. They argued that the response of mosquitoes to different repellents, and presumably attractants, is species-specific [4]. We would add that each sex of a mosquito species might also respond differently to odour stimuli, as observed, for example, with foot odour.

We found that geranyl acetate elicited no response at any concentration, however, repellent activity assays with female *Ae. aegypti* found that higher concentrations (10 and 25%) offered 97–100% protection for more than 60 min [53].

The compound, 3,7-dimethyl-oct-6-en-1-yn-3-ol (dehydrolinalool), which is present in the floral odours of many plant species, elicited no response from male *Ae. albopictus*, except at the highest concentration, when it acted as a repellent. For *Ae. aegypti*, however, it resulted in activation and/or orientation towards the chemical source [54].

### 4.3. Putative Pheromones

Several laboratory studies have shown that swarming *Ae. aegypti* males or females upwind in an olfactometer elicit a flight response in female mosquitoes, suggesting that both sexes produce volatile signals that can initiate swarm formation [10]. Indeed, two of the compounds that we found to be repellent at the highest concentration, 2,6,6-trimethylcyclohex-2-ene-1,4-dione and 4′-ethyl-acetophenone, were isolated and identified as aggregation pheromones for *Ae. aegypti* together with another untested compound, 2,2,6-trimethylcyclohexane-1,4-dione, [12]. 2,6,6-trimethylcyclohex-2-ene-1,4-dione and 2,2,6-trimethylcyclohexane-1,4-dione elicited a flight excitation effect on females, whereas 4′-ethyl-acetophenone acted as an attractant at a concentration comparable to our intermediate concentration. Male *Ae. aegypti* responded only to 2,6,6-trimethylcyclohex-2-ene-1,4-dione, in a dose-dependent manner, with a characteristic flight pattern similar to swarming, where the number of males participating in the swarm was a function of the concentration of the compound [12].

## 5. Conclusions

Some considerations emerge from these preliminary data. Firstly, compounds that are emitted by plants and animals may elicit different responses depending on its concentration and the species and sex of the mosquito. Moreover, the effect of a single compound may be enhanced or diminished, depending on the other compounds that are present. Further investigations should evaluate different blends of these compounds, as many studies have shown that blends are more likely to elicit a response than individual volatiles [55,56]. Wide-scale tests should be performed with individual candidate compounds, and mixtures of compounds, in the environment where the mosquito is found and where trapping is planned, testing different concentrations, formulations and methods of emission. These wide-scale tests could be performed in association with an evironmental evaluation of the presence of plants, animals and fungi to help explain the absence of mosquitoes and frequent negative collections at certain trapping points.

One of the compounds tested in this study, decanoic acid, appears to be a promising candidate for male trapping. Further tests in the field, perhaps in combination with other compounds or sound stimuli [57], will be necessary to identify the optimal dose for its potential use in monitoring during SIT and population-modification programmes aimed at *Ae. albopictus* populations.

## Figures and Tables

**Figure 1 insects-13-00290-f001:**
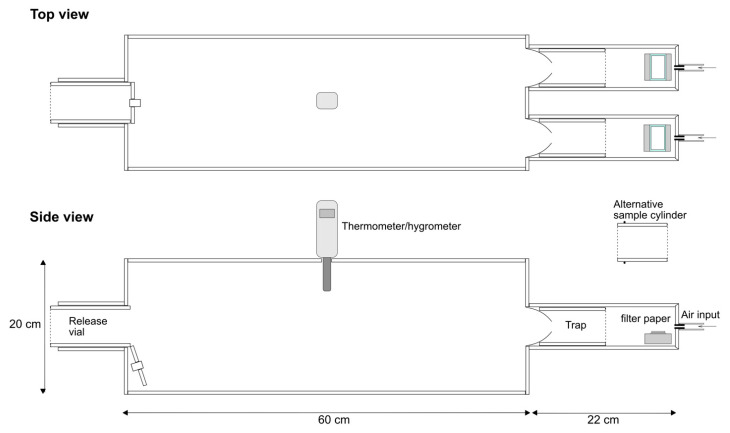
Schematic diagram of the dual-port olfactometer.

**Figure 2 insects-13-00290-f002:**
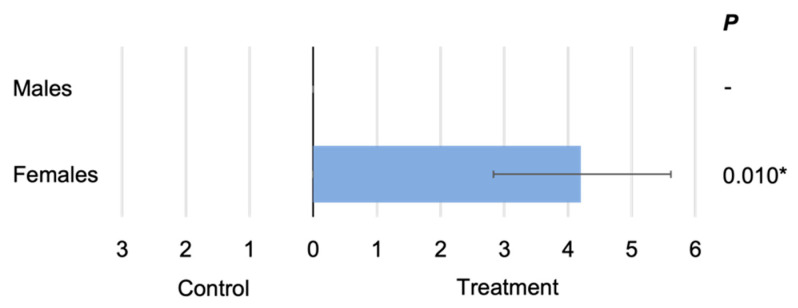
Male and female responses to worn and control socks. The blue bar represents the mean number (±SE) of individuals trapped in the test (worn sock) or control chamber. The sex of the tested mosquitoes is indicated on the left of the graph and the associated paired, two tailed *t* test *p* values are on the right (* *p* < 0.05).

**Figure 3 insects-13-00290-f003:**
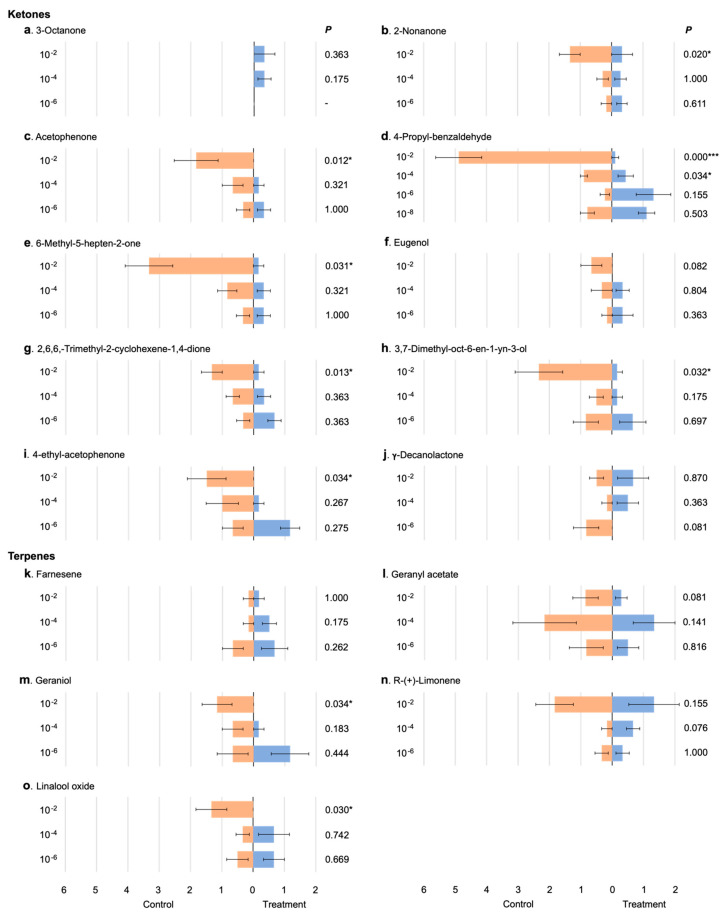
Responses of males (and females for decanoic acid) to different concentrations of chemical compounds. The orange and blue bars represent the mean number (±SE) of individuals trapped in the control and test chambers, respectively. The concentrations of the test compound are indicated on the left of each graph and the associated paired, two tailed *t* test *p* values are on the right (* *p* < 0.05; ** *p* < 0.01, *** *p* < 0.001).

**Table 1 insects-13-00290-t001:** Chemical compounds used in this study.

Compound	Sigma-Aldrich Product No.	Purity ^1^	Family	EC No. ^2^	Origin: Perceived Odour ^3^
3-Octanone	136913	>98%	Ketone	203-423-0	Fungal: fresh, herbal, lavender, mushroom
2-Nonanone	108731	>99%	Ketone	212-480-0	Animal: fragrant, fruit, green, hot milk
Acetophenone	42163	NS	Ketone	202-708-7	Plant: almond, floral, sweet, cherry
4-Propyl-benzaldehyde	562882	95%	Ketone	249-221-6	-
6-Methyl-5-hepten-2-one (sulcatone)	M48805	>98%	Ketone	203-816-7	Animal: citrus, green, musty
Eugenol	E51791	99%	Ketone	202-589-1	Plant: clove
2,6,6-Trimethyl-2-cyclohexene-1,4-dione	14239	≥98%	Ketone	214-406-2	Pheromone: sweet, leaf, floral, tobacco
3,7-Dimethyl-oct-6-en-1-yn-3-ol (Dehydrolinalool)	CDS009913	≤100%	Ketone	249-482-6	Plant: mould
4′-Ethyl-acetophenone	226750	97%	Ketone	213-326-5	Pheromone: floral, hawthorn
γ-Decanolactone	D804	98%	Ketone	211-892-8	Plant: fat, fruit, lactone, peach
Farnesene, mix of isomers	W383902	NS	Terpene	207-948-6	Plant: green apple
Geranyl acetate	173495	≥97%	Terpene	203-341-5	Plant: floral
Geraniol	163333	98%	Terpene	203-377-1	Plant: geranium, lemon peel, passion fruit, peach, rose
R-(+)-limonene	183164	97%	Terpene	227-813-5	Plant: citrus
Linalool oxide	62141	≥97%	Terpene	262-038-6	Plant: floral, herbal
Hexyl hexanoate	W257206	≥97%	Ester	228-952-4	Plant: apple peel, peach, plum
Hexyl-2-methylbutanoate	W349909	≥95%	Ester	233-106-2	Plant: sweet, green apple
Methyl benzoate	M29908	99%	Ester	202-259-7	Plant: phenolic, almond, floral
Methyl 2-methyl benzoate	259985	99%	Ester	201-932-2	Plant: floral, orange flower
Methyl salicylate	M6752	≥99%	Ester	204-317-7	Plant: liniments, hospital
Methyl cinnamate	96410	≥99%	Ester	203-093-8	Plant: balsamic, cinnamon
*cis*-3-Hexenyl acetate	W317101	≥98%	Ester	222-960-1	Plant: banana, candy, floral, green
Nonanal	W278220	97%	Aldehyde	204-688-5	Animal/Plant: fat, rose, orange
Phenylacetaldehyde	W287407	≥95%	Aldehyde	204-574-5	Plant: honey, sweet, rose, green
*p*-Anisaldehyde	A88107	98%	Aldehyde	204-602-6	Plant: floral, aniseed
2-Phenyl ethanol	77861	≥99%	Alcohol	200-456-2	Plant: floral, honey, rose
1-Octen-3-ol	O5284	98%	Alcohol	222-226-0	Animal: mushroom, earthy
Decanoic acid	W236403	≥98%	Fatty acid	206-376-4	Animal: goat, fat, grass, dust

^1^ NS: not stated by supplier. ^2^ European Community number. ^3^ Derived from PubChem [21] and TGSC Information System [22].

## Data Availability

The data present in this study are available in the Appendix A.

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
