# Peer review of "Behavioural Responses of Male *Aedes albopictus* to Different Volatile Chemical Compounds"

_insects, 2022, doi:10.3390/insects13030290_

Round 1

Reviewer 1 Report

Notwithstanding the manuscript is well written and deals with an interesting topic, there are several weaknesses.

  • The manuscript is missing of the rationale that explain why the authors started with testing human odor as stimuli for mosquito attraction and then, without any chemical analysis of the odor blend they test a series of chemicals coming from different host/substrates. If, as I understand from the discussion section, the bioassay with the sock was done just for validating the olfactometer, this should be stated clearly in MM section.
  • Statistics are not convincing me as graphically some response look very low and is reported as statistically significant.
  • The number of replication for such type of behavioral observation is too low, I’d use less chemicals or less doses but with more replications (at least 10).
  • I argue that in this type of bioassays the repellency can be established.
  • Results are poor, the only compound eliciting attraction, is totally unattractive at the higher dose, or as the author write “repellent”.

L43, add reference

L52, add reference

L61, “animal host emanations” does not make clear that is a human host odor, I suggest to give more detail.

L72, if available give the detail of the bugdorm furnisher.

L93,” volunteer (LMG)” what does LMG stand for? Was the volunteer a man or a woman?

L109, five replications for such experiment is a quite low number.

Line 116, give details of the supplier.

Line 146, I argue that repellency can be assessed like this.

Figure 2. is very unclear. What does the bar represent? Are these the arcsin transformed values?

Figure 3 caption should be put together. Again the responses are not clear, and in some case I doubt that the statistics are valid, once the responses are very low but still significant, see for example the response to nonanal.

Line 235-237 the

L201-202 If “Our aim was to determine the responses of male Ae. albopictus to compounds that are present in volatiles emitted by plants, fungi or animals” it should be carefully explained why 1- the authors tested the socks and 2- why they choose such chemicals. If 1 was only done for validate the olfactometer it should be more clearly reported in the MM section.

L 205-207, this should go in MM section

L210-215, delete

L235-237 the color of the text is different

L318-319, hard to understand

Reviewer 2 Report

Behavioural responses of male Aedes albopictus to different volatile chemical compounds

Manuscript ID: insects-1576107

The article is well-written, I have only a few questions.

L64 - please put the hypothesis in the end of Introduction section

L73 – what was the ratio female:male

Figures and Tables – please add information about statistical test you used to compare data

L316 – it is worth to mention that decanoic acid is also part of mosquito cuticular and internal lipids, and its amount is depended on method of feeding (Kaczmarek A, Wrońska AK, Boguś MI, Kazek M, Gliniewicz A, Mikulak E, et al. (2021) The type of blood used to feed Aedes aegypti females affects their cuticular and internal free fatty acid (FFA) profiles. PLoS ONE 16(4): e0251100. https://doi.org/10.1371/journal.pone.0251100)

Reviewer 3 Report

I was really driven by the abstract and simple summary. But my expectations as a reader were not fully met. Following are the major and minor criticisms of the manuscript:

L15: Nowhere in the manuscript authors mention why 28 compounds were chosen? There needs to be a reason why.

L18: Authors mentioned that they would test components of Aedes aegypti but nowhere in the manuscript I saw the reason why they were selected for testing.

L43: Reference missing.

L47: Please make sure the latin names are written correctly.

L49: Authors speak of 3 species in the previous sentence, so in the preceding sentence it is not clear which species they refer?

L48-56: I find it difficult to find connection between sentences. This part needs to be rephrased to make it more clear to understand.

L61: Part of introduction suggests that authors will test plants. But they never mentioned before that they would test volatiles from different sources. These things should be uniform throughout.

L99: Would be ideal to specify the photoperiod in lux just above the olfactometer surface. This will give readers an idea how much of light mosquitoes received during the experiment.

L109: Please rephrase the sentence.

L112: Please specify how this time window overlay with the photoperiod.

L119: Please help the readers by explaining why these concentrations were tested.

L121: it’s quite unusual to take 400 uL of hexane, and then wait for 2 minutes. Authors loose considerable amount of compounds during this period. This will affect any conclusions coming out for low-concentration compounds. Why authors did not standardize the amount of compound to be tested by taking less volume of hexane?

L125: typo “solid glass watch glass”.

L143: Please explain why this time window? Are there any reference to show why this time window is important compared to, for example, early morning hours?

L145: “in terms of attraction and repulsion”

Table1: I recommend that the authors mention references for “Origin: perceived odour”. Without references, they seem off. What is EC No. Do you mean CAS number? Further, I don’t understand why “:” are used (Example: Animal: fat, rose, orange) for Nonanal.

Fig2: I see stats for males in the figure. But in the text, it suggests other way around. The stats for males are not shown here. Why was this experiment important? If the authors wanted to test if their olfactometer setup is working, they could have simply tested know attractant / repellent, for example Deet. Testing a sock here is not in line with this manuscript because authors intended to test synthetic compounds. The study is about floral volatiles and the responses in males. The behavioral experiments are not backed by GC-MS, or any quantitative analysis. So, I don’t see section 3.1 as part of this study.

L120: Only until 10^-6 is shown in the material and methods, and not 10^-8.

L188: Why females were tested only for acid? First of all, the manuscript (by the title) is regarding males and not females.

L217-223: Part of the introduction and not discussion.

L223: Elciting antennal response do not tell us if these compounds were attractants or repellents.

L217: “Mosquitoes visit flowers for nectar”. “The flowers of many plants are visited by” is passive.

L217-231: The results are repeated here. I don’t see this as part of discussion.

Overall, the discussion is not up to the mark, and must be rephrased / re-written in several parts of the discussion.

L248: The cited article work is not on tick or black flies. Authors must be careful how they cite other works.

L257-260: Not relevant.

I suggest authors to get help from native English speaker for grammatical errors.

The discussion did not need to be this big by comparing other works (compound b by compound). I see that authors are trying to show references to each compound they have tested to make their results relevant. But I am afraid majority of the discussion is redundant.

Reviewer 4 Report

The Asian tiger mosquito, Aedes albopictus, is an important invasive vector of disease pathogens such as the viruses that cause chikungunya and dengue.  Male individual of this kind of mosquito require nectar as a source of carbohydrates and respond to semiochemicals released by suitable host plants. This paper reported the responses of male Ae. albopictus to 59 different concentrations of twenty-eight chemical compounds that are present in volatiles emitted by plants, fungi or animals after indoor test with a dual-port olfactometer. They found that 13 acted as repellents at one or more concentrations, fourteen compounds elicited no significant response, and one compound (decanoic acid) acted as an attractant to males at the intermediate concentration. These results are useful to develop trapping technology for male mosquitoes of Ae. albopictus. However, the present data are from the preliminary tests at three concentrations of compound: 1%, 0.01% and 0.0001%. More concentrations (such as 0.05%, 0.005%) need to be tested. If that, we can get more important information about compounds to male mosquitoes. In addition, why did authors use six to ten day old male and female Ae. albopictus individuals for tests? Normally, newly emerged adults are better for monitor purpose in the field.

Round 2

Reviewer 1 Report

After the Reviewers’ comments, the authors partially improved this manuscript, that deals with a very difficult and complex task.

However, as they argue with some of my comments i.e. that the number of replicates is not too low and that the instrument used (a 2-choice olfactometer) is valid to estimate repellent compounds, and they think that the data reported are not poor, some distance between our point of view remains.

In this research, was not found any real promising candidate attractant for males, and the authors pointed out that this was a main aim of the work as in simple summary they state that “The identification of chemical substances that attract males can be very useful for trapping purposes…” and in the objectives that “Our hypothesis was that male mosquitoes may be attracted to one or more concentrations of these compounds.”

Few other issues:

If the bioassays with the sock was done to validate the instrument, and the males didn’t respond at all, the validation is true only for the female mosquitoes… and the authors used males mainly.

Another concern is why the data in the figures are showed as “arcsin transformation” this is not giving any real information. For statistics, the author can indeed transform the data (if they are NOT normal, and this should be pointed out) but in the graphs it should be reported the mean of the responses or the percentages of choices.

In the discussion the authors should focus mainly on decanoic acid the only one that elicited some weak attraction and minimize the discussion related to other stimuli as they were not effective or often even repellents.

L90 “…material such as socks (see below).” Please change this sentence, just specify that could contain an attractant stimulus.

L96, change in “olfactometer validation”, similarly in the results section

L134, males of females?

L140, individuals are only males?

L154-156 unclear, suggest deleting.

L171, if the main goal was to demonstrate that some compound was attractant to use it to monitoring males, the author should focus on that first. Moreover, I suggest to remind here that only males were tested.

L197-198, only for this compound? In case specify

L220, I suggest to delete “solvents” as they should evaporate before experiments.

L366-367, this statement has been observed several times in entomology.

Reviewer 3 Report

Following are some of the major and minor concerns

L18: In m&m, you mention animals, plants and microbes?

L102: Distal

L134-142: Repetitive. You already mentioned this for ‘sock’ assay.

L202: It is fair to provide why authors decided to transform the data before analysis. You could have also done this analysis using actual experimental outcome numbers. Do consider to provide n values in the paranthesis for all the graphs.

Discussion:

All I hoped was that authors would reconstruct their discussion to ‘discuss’ the bigger picture of their results. First of all, it is not clear in the entire manuscript, why authors chose 28 compounds representing different chemical classes. It could have been 100 compounds, with known behavioral and electrophysiologically-active compounds. So basically, I lost my interest in reading the manuscript when I noticed that there still wasn’t any rationale mentioned to say why these compounds and why only 28 compounds were chosen.

Authors initially tested females to setup their olfactometer, and then chose males for their experiment. Throughout the discussion, they switch back and forth to compare their results to males and females of other mosquito species. Even more, they compared their results to behavioral responses in hoverflies (Diptera), moths (Lepidoptera), beetles (Coleoptera), and even ticks. The authors have only one interesting acid compound, that could further their research by including more behavioral data and both sexes, but also include chemical analysis (GC-MS) data to show where they find this acid, and how is it relevant for the entire study.

Basically, authors have picked all articles that show any behavioral responses (attraction/repulsion/none) to the compounds they chose to test, and then they have concluded that they also found responses in male mosquitoes. Dividing the discussion based on chemicals, and then not having a bigger picture is the major concern of this manuscript.

Reviewer 4 Report

The authors have made better revision on the manuscript, it can be accepted to be published.

Author Response

We thank the reviewer for his comment.

Round 3

Reviewer 1 Report

The paper reports a very unlucky experiment that is dealing with a very complex task as the development of a semiochemical-based attractant for mosquitoes.

The results were negative as almost none of the compounds tested can be considered a promising attractant. Furthermore, once again I have to state that the rationale of the experiment is unclear.

The authors neglected or disagree with several comments of my past review.

Again, I think that the number of replications is pretty low for such kind of experiment, so I suggested to move a step backward in the lab and at least arrive to 8-10 replications per treatment.

The fact that the author validated the experiment for attraction response and later on write about repellent response elicited by several chemicals is misleading.

The discussion of the results has been changed but still must be improved (rewritten). Sometimes the sentences don’t have a rationale (see line 247) or misleading (see lines 312-314). The authors should focus at the beginning of the section on which are the main results obtained and compare this with other studies… Another important aspect that should be underlined is that in the majority of cases is a blend of ubiquitary compound that elicit an attraction response for the insects feeding and ovipostion substrates, see for example Bruce, T. J., & Pickett, J. A. (2011). Perception of plant volatile blends by herbivorous insects–finding the right mix. Phytochemistry72(13), 1605-1611 and much more rarely the attraction is determined by a single compound, this could be a reason why the experiment failed. This was only briefly addressed.

Furthermore, the authors in discussion neglected some new advances on mosquito attractants (also A. albopictus), see for example

Dormont, L., Mulatier, M., Carrasco, D., & Cohuet, A. (2021). Mosquito attractants. Journal of chemical ecology47(4), 351-393.

Nyasembe, V. O., & Torto, B. (2014). Volatile phytochemicals as mosquito semiochemicals. Phytochemistry letters8, 196-201.

Díaz-Santiz, E., Rojas, J. C., Casas-Martínez, M., Cruz-López, L., & Malo, E. A. (2020). Rat volatiles as an attractant source for the Asian tiger mosquito, Aedes albopictus. Scientific reports10(1), 1-12.

L222-224, this is not discussion is introduction (objectives already stated)

L224-226 this is part of materials and methods, not discussion.

L242-244, this was stated above

L247, “Decanoic acid is also an efficient larvicide at 150 ppm when applied to pools” what this has to do with your results (weak attraction)?

Line 254, “showed repellency” (Here and across the manuscript) is not correct because the olfactometer was validated for attraction not for repellency.

L311, “These data suggest that the concentration of a chemical (and assay type) may influence the behavioural response” this aspect has been observed hundreds times in literature. Moreover, what the authors mean with “assay type”?

L312-314, this was observed ONLY for decanoic acid.

Reviewer 3 Report

I believe the authors have made efforts to improve the manuscript, and I am satisfied with their efforts. Although, there is always room for improvement in the language throughout the manuscript.

Following are some of minor concerns

L50: mosquito species

L52: Certain host cues, such as carbon di oxide however….

L133: what is the thickness of the filter paper?

L169-172: Your results defy that is written in L53 and L54. How do you explain these results?

L173: Which of these figures is the ‘right’ one? Please do mention somewhere in your explanation what is on the X-axis for your graphs? Is it Preference index or something else(?)

L292: between species

L312: “Higher concentrations are often….” This is not true. Please erase this statement.
